# Analysis of the Factors That Influence a Quality Physical Education in Mexico: School Supervision’s Perspective

**DOI:** 10.3390/ijerph19052869

**Published:** 2022-03-01

**Authors:** Ramón Alfonso González-Rivas, Oscar Núñez Enríquez, María del Carmen Zueck-Enríquez, Gabriel Gastelum-Cuadras, Adrián Alonzo Ramírez-García, Salvador Jesús López-Alonzo, Julio César Guedea-Delgado

**Affiliations:** 1Faculty of Physical Culture Sciences, Autonomous University of Chihuahua, Chihuahua 31000, Mexico; rgrivas@uach.mx (R.A.G.-R.); mzueck@uach.mx (M.d.C.Z.-E.); gastelum@uach.mx (G.G.-C.); dr.adrianramirezgarcia@gmail.com (A.A.R.-G.); salopez@uach.mx (S.J.L.-A.); jcguedea@uach.mx (J.C.G.-D.); 2Physical Education Department, Chihuahua State Educational Services, Chihuahua 31210, Mexico

**Keywords:** physical education, public education, educational policy, educational organization

## Abstract

Different international organizations have spoken out in favor of the value of physical activity in the integral development of children and adolescents, which is why different guidelines have been established for governments to facilitate quality Physical Education. In this sense, the objective of this study was to analyze the factors that influence the performance of PE teachers, from the perspective of pedagogical technical advisors and school district supervisors from various school zones from the State of Chihuahua. This was a transversal study, with a qualitative approach and phenomenological focus, which was chosen because it allows the understanding of these aspects from a unique and unrepeatable process; data collection was carried out through focus groups in which PE ATP supervisors participated and data were collaboratively analyzed. The main results indicated that Physical Education does not have the same importance as other subjects, a situation that is reflected in: constant curricular reforms, administrative inconsistencies, dependence on the school organization, and being seldom taken into account in decision-making. It was concluded that strict public policies are required to be applied uniformly throughout the country, as well as privilege Physical Education within the Basic Education curriculum in Mexico.

## 1. Introduction

According to the World Health Organization [1], physical activity can improve the quality of life and contributes to preventing illnesses such as hypertension and diabetes. However, it seems that the practice levels of physical activity around the world have decreased significantly [2]. In this way, the United Nations Educational, Scientific and Cultural Organization (UNESCO) [3] recognizes the importance of physical education (PE) services as the first structured space where children and youth can engage in physical activity, and thus invites government institutions and departments to develop and implement educational policies that contribute to the promotion of and engagement in a quality PE service. In addition, the International Physical Education Federation (FIEP) [4] establishes that educational authorities and governments should be in favor of strengthening policies that promote PE services, especially in socially vulnerable contexts. 

In Mexico, the mission of the Public Education Department (SEP), which is the educational governing body, is to guarantee a quality education to all of the Mexican population through its educational plan of basic education, which includes kindergarten through ninth grade (middle school), considering PE services as a mandatory subject [5]. This plan recognizes and follows the WHO guidelines about the importance of PE to prevent nontransmissible diseases. As such, the SEP created the Quality Physical Activity Technical Group whose function is to guarantee a quality PE [6]. According to Martínez et al [7] it is important to maintain government efforts to guarantee a quality PE that includes the value to consider these educational policies in benefit of the surrounding community. 

The SEP’s [8] education plan named Key Learning aspects establishes the guidelines to promote a quality PE service in Mexico, and for the first time considers the equity and inclusion of special populations. Among its objectives is the seeking of guaranteeing that all children and youth have free access to education, with the goal to learn relevant skills and knowledge for life. In this way, such a program refers to the need to improve the holistic development among students and use a pedagogical approach that lets teachers and students work together. 

However, the current literature mentions that Mexico lacks strict policies that contribute to the full engagement of physical activity in the school-age population [9], despite what the education plans and educational system establishes. One reason is the educational policies’ inconsistences that affect the decision-making process between authorities and teachers. Among the decision that authorities and teachers need to take are the administration, curriculum, and equipment/facilities, which have repercussions in accomplishing a quality PE service, thus affecting the acquisition of healthy habits among students [10,11,12,13,14]. In addition, the school individual policies may sometimes create a struggle for teachers, affecting them and students as well [15].

A consequence of this issue is that, in Mexico, 36.3% of students from public schools have PE only one hour per week [11], although 34.8% of these students are physically active in their free time, and only 12% engage in a high level of physical activity on a daily basis [16]. This is congruent with the low perception of PE and the lower level of satisfaction of PE compared to Spain [17]. At the same time, in Mexico, there is an alarming sedentary lifestyle and inactivity among children and youth [7,9], especially in northern parts of the country; there is a high incidence of students with overweight and obesity, which is correlated with the lack and null practice of physical activity per week. Thus, it is essential to promote physical education spaces that encourage children and youth to seek their motivation, interest, and learning to engage in physical activity for a lifetime [18,19,20].

Within the Mexican education system, the PE subject holds three key figures that support the pedagogy and administrative nature of the classes, having a direct impact on students. The first is the teacher, who has a direct impact on students. The second is known as the Pedagogical Technical Advisor (ATP), which is the link between teachers and the administrative and professional development aspects of the education system. The last one is the link between the educational system and the teachers, because it helps teachers and the ATP have a complete vision of the issues that, between both ends, may also demonstrate how to confront them. This figure is known as the district supervisor. It is important to say that the Mexican educational system is divided by districts or school zones that hold a specific number of schools and teachers. The difference between other educational systems around the world is that Mexican PE teachers tend to teach in a minimum of two schools per week; thus, their teaching can vary from schools, grades, days, and school zones too.

As it was mentioned, the SEP recognizes the importance of PE in Mexico; however, there is background that indicates the deficiencies that directly affect classes and the guidelines proposed by the education system. In this way, the purpose of this project was to understand the reality of the PE organization and to analyze the factors that influence the PE teacher’s performance from the perspective of the ATP and district supervisors, because these two figures are the link between administrative authorities and the reality that teachers face on a daily basis. Thus, understanding these points of view can create a link between policies, theory, and practice that PE requires nowadays.

## 2. Materials and Methods

### 2.1. Research Design

This was a transversal study, with a qualitative approach and phenomenological focus, which was chosen because it allows the understanding of these aspects from a unique and unrepeatable process, because it takes personal and collective experiences to build a construct that shapes the reality [21,22]. In addition, the Helsinki declaration [23] guidelines were considered.

### 2.2. Participants

This was the result of a larger research that involved different PE actors, such as ATPs and Supervisors. We wanted to understand their thoughts about their view of PE classes. We also wanted to highlight and comprehend the supervisor’s perspective, because, from this view, there are several educational policies that can take place, thus affecting the PE class climate in a different school district. As such, the inclusion criteria for this project focused on the school supervisors’ perspective. Participants needed to have: (1) at least 2 years of experience as supervisor; (2) at least 20 PE teachers under his/her supervision; (3) a graduate degree; and (4) a willingness to participate and share their opinion openly. Exclusion criteria were not attending the first focus group meeting. We provide a description of the three participants of this study below:Joel, a 41-year-old male, has 20 years of teaching experience. He is the supervisor of a rural school district, whose location is in the southeast part of the state. He holds a PE bachelor’s and master’s degree. As school supervisor, he holds 3 years of experience and supervises 83 schools (K-6) and a total 38 PE teachers.Monica, a 38-year-old female, has 15 years of teaching experience. She is the supervisor of one of the most extensive urban school districts in the state, because her school district borders with the U.S. She holds a PE bachelor’s and master’s degree, with 6 years as supervisor, where she supervises 50 schools (K-6) and a total of 28 teachers.Esteban is a 44-year-old male, with 22 years of teaching experience. He is the supervisor of a rural school district, which is located in the southwest part of the state, being the most extensive district as it is surrounded with the sierra madre. He holds a PE bachelor’s and master’s degree, with 3 years of experience as supervisor in which he supervises 50 schools and a total of 28 teachers.

### 2.3. Instruments

Through focus groups, interview data were collected. Kitzinger [24] defined focus group interviews as an in-depth interview that uses groups and a moderator to do so. It also allows participants to interact but gives interviewees full leadership; as such, it allows the collection of data that, with other sources, would be difficult to do so [25]. Hamui-Sutton and Varela-Ruiz [26] mentioned that for a focus group to be successful, the selection of the participants is important.

For the data analysis, the Atlas Ti software was used, which is a tool and instrument that facilitates the coding process; however, the way these data are coded by the researcher is important [27]. In addition, this tool has been used in diverse qualitative research projects, which has a great potential of analysis [28].

### 2.4. Procedure and Data Analysis

First, the project was approved by the revision committee from the Autonomous University of Chihuahua (Registration and Approval number 22032021-017). Once authorization was obtained, the project was presented to the State of Chihuahua Educational Services (SEECH). After presenting the project, we gathered the research group in order to design specific research questions in 5 meetings of 90 min each, for the specific moments of the PE classes. An open invitation to ATPs and Supervisors was made; however, there were five ATPs and five Supervisors selected. There were three focus group meetings; however, to these meetings, only three supervisors attended. These meetings were held in April, 2021 simultaneously through the ZOOM platform. These were recorded and transcribed verbatim for further analysis with the ATLAS ti software (Scientific Software Development, Berlin, Germany). Through these meetings, a current topic that was pointed out referred to the educational policies, organization, school, and class climate having an influence on the teacher’s performance. Thus, this project only presents the experiences of three PE supervisors.

The data analysis was collaborative [29], which means that the research group delimited the object of the study based on a referential framework. According to Richards and Hemphill [30], a collaborative qualitative data analysis is based on a series of steps in order to create a strategy to enhance the trustworthiness of the process: (1) Preliminary organization and planning, (2) open and axial coding, (3) development of a preliminary codebook, (4) pilot-testing the codebook, (5) final coding process, and (6) reviewing the codebook and finalizing the themes. Once these series of steps were completed, the research group created a coding process individually. The following step was to establish, create, and agree to a coding book where all researchers’ coding was processed through a collaborative reflection and final coding process, in order to establish the codebook and themes (e.g., lack of value of PE, planning process, school management, school organization dependance, and not taken into account). As a way to strengthen the trustworthiness of the collaborative process, a second data analysis was performed, including the agreed codebook, and analyzing the data again. In this way, the theme structure was generated using the systematic design, that was established as a research group, which strengthens the data analysis and coding [29,30].

## 3. Results

Once all data were analyzed, it was possible to identify diverse factors that influence the PE class. Thus, we describe the impact of these factors and how these were represented in the following findings:

### 3.1. Lack of Value of PE

According to the supervisors, PE was not given the same importance compared to other subjects; in addition, there are other factors that can hinder the quality of PE such as the constant changes in the plans and programs throughout the years, administrative incongruences, school organization dependency, and being a subject not valued or considered in the school decision-making process. These aspects together have damaged the design, implementation, and following projects in PE, which seems to prevail nowadays:

…I’d like to go back in time where there was the improvement route plan and the focus was in health, so teachers started to plan their classes with a health focus, considering the school lunch, and other aspects that schools had at the time. Unfortunately, these projects, couldn’t accomplish anything, couldn’t move forward, because results were asked, these results weren’t accomplished because some schools didn’t care, they cared more about Spanish and Math, so it was nothing else beyond this scope, which discourage some teachers. …(Mónica)

### 3.2. Planning Process: Constant PE Programs Changes

From the supervisor’s perspective, the educational system planning process has opportunities to improve areas that should be taken into consideration to offer a quality PE class. In recent years, there has been a series of educational reforms that has not allowed teachers to learn, understand, and implement fully the current PE program, which has caused teachers to limit their pedagogical planning; thus, teachers tend to use only activities that they feel comfortable with or can remember from previous programs. At the same time, beginning teachers have difficulties and the education system does not offer any support of professional development. As such, supervisors recognize that PE faces a constant change and evolution, but to guarantee a better understanding and teacher candidates, it is fundamental to establish a link between school districts and universities to retain updated knowledge to all participants.

…mostly on the programs, after the dynamic motor integration program, the expected learning guidelines are too long to be accomplished, before they were easy to implement and understand, to understand the concept and knowing how to get use. But now, in one expected learning guideline are too many elements that were used before but separately, which created a lot of confusing among teachers…(Mónica)

…The evaluation, the practice, planning processes, because we’re here to know the competence situations, we have to support teacher to let them learn to learn, to have the will to learn, because it doesn’t want to learn for her/himself, the system would not help him/her …(Joel)

…as for the professional development for teachers, how are students graduating from the university? What attitude do they have? What competences are they graduating with to apply in a teaching job? Perhaps they have memorized the program, but if they are not competent enough to analyze, reflect, synthesize about all these aspects, then we can begin to move forward …(Joel)

### 3.3. School Management

In addition, there were organization and administrative aspects mentioned, which impact negatively on teachers’ performance; these aspects are related to the type of hiring process as PE teachers, the lack of support to obtain/purchase equipment and infrastructure in some schools, and the excessive number of students in class. All these factors combined had affected the chance to offer a quality PE class.

… How can be possible that there are teachers hired in one way and other in other way, this creates a first incongruency, this affects having a lot of students, groups, and creates this vicious cicle to go over and over again …(Joel)

… to contextualize there are some schools that have the means and resources to buy equipment…(Esteban)

### 3.4. School Organization Dependence

According to the PE supervisors, teachers depend on the educational guidelines and policies; however, each has their own policies, resources, and infrastructure, which are sometimes split within the school district. In this way, teachers are constantly in need to adapt themselves to each particular situation and each school. Thus, it is complicated to implement a specific PE project within the same school district, which has created discomfort and frustration with the supervisors.

…So, depending from a third party in organization within schools, there’s where we realice the lack or null support that the education system offers nationwide… even in regular times we depend from certain decisions of school’s organization, imagine nowadays…(Joel)

…Well, I don’t like it! That dependency that we have, in school organization, what would be do? well to balance each teacher’s situation. It bothers me because is nearly impossible to create a school district project and being implemented as it was planned. There are a lot to be done, but once it reaches the school changes need to be made, to fit the school needs because each are different…(Mónica)

### 3.5. Not to Be Taken into Account

Lastly, and following with the lack or null consideration and value of PE, it was also raised that PE teachers are not taken into this situation, especially in the schools’ decision-making process. In this way, supervisors mentioned having the will to be taken into consideration in major decision-making that affects the schools’ organization.

…information comes out and it goes directly to the schools, but there aren’t any for us, we’re always creative and looking how to deal with it …(Esteban)

…To me, I can conclude that we have that expectancy because things are not clear enough, we need to participate in such decision-making process, to speak-up, to ask, to collaborative work so we can be taken into consideration…(Joel)

… But we also have high expectancies in our education authorities, to get their act together, and together with us in these informative and feedback meetings, where we can work together and create professional development courses. I really hope so, because it is sad to see that we’re lost, scattered, everyone by their own, it is sad to see everyone asking someone else, asking secretly to the supervision and other people…(Mónica)

As such it is important to say that these aspects had a stronger connection with each other, as the impact from each one had an indirect and direct impact on the others. For instance, the different changes in the planning process have a connection with the little value that PE has had over the years, thus creating a lack of perspective from teachers in the educational system. The school management tends to have a direct impact on the lack of or null equipment resources for PE, just to mention the connection between a few of the elements. These aspects and its connection are displayed in Figure 1:

## 4. Discussion

The purpose of this project was to analyze the factors that influence PE teachers’ performance, from the perspective of ATPs and school district supervisors. The importance of this project falls under providing current information that identifies the reality that Mexican PE teachers face in order to contribute to a quality PE class. Findings in this project indicate that PE in Mexico faces diverse administrative, political, and organization issues. Examples of these issues can be described through the several educational reforms, programs and curriculum changes, hiring processes, lack of support to novice teachers, lack of resources and facilities, and lack of support and involvement in the decision-making process that some school require, giving null or little value to the PE subject, as shown in Figure 1.

One finding of this project was that the PE subject has a lack of support and importance by educational authorities. This situation is evident in the educational curriculum and in the lack of value in PE projects per school, because the PE teacher falls under the school district supervisor; however, the school principal carries an indirect authority and decision process that can hold the PE process. This devaluation has also been identified by an international organization, where teachers had been found to have a different status than other teachers [31]. These findings are in the same line of research than García and Del Basto [32], which highlight the importance of improving the PE curricula due to the increased number of children and youth with obesity and overweight in Mexico. However, this is not a situation that Mexico faces solely, as the United States has established guidelines in favor of PE, but not all schools abide by these guidelines [32,33]. For PE to accomplish the goal to create healthy habits for a lifetime among children and youth, it is essential to create strict educational policies that support PE teachers in having a major presence within the education curriculum, focusing on the importance of being physically active and using an appropriate approach [18,34,35].

In this project, it was found that the constant changes including educational reforms had generated confusion among PE teachers, having a direct effect on teachers’ practice because they are, in different ways, forced to create and design strategies into their teaching abilities. Garcia and Hoil [14] found that curriculum changes are the result of political and union factors. These results suggest a lack of understanding and knowledge of the international recommendations, which suggest PE as the ideal way to create change [10]. At the same time, this has brought the opportunity to understand the effect of school policy changes on teaching, teachers, and students in order to diminish the current gap between them. In this way, PE must transcend through governments and not be bound to temporary solutions. Thus, it is recommended to create a stronger link between higher education institutions and schools, in order to connect theory, practice, research, and reality to increase students’ engagement.

According to the results presented by Olvera et al. [13], in Mexico, there are certain school districts that implement only three hours of PE per week, other districts that implement only two, and those in the southern part of country that implement only one hour per week. The UNESCO [30] mentions that Mexico is one of the Latin-American countries that provides less time to PE classes either in rural or urban areas, and the number of students is a factor that influences the number of hours to be given in classes. It is important to mention that PE is the only place in which some children and youth engage in physical activity because it is a mandatory subject [36]; as such, the classes must accomplish a lifelong effect to acquire healthy habits among students. This effect can indicate the lack of school policies that provide curriculum guidelines to unify the amount of time given to PE per week nationwide in order to let children and youth engage in the daily required amounts of physical activity. However, this imbalance in the amount of time per week influences the development of nationwide projects that involve PE, as it seems that three classes of PE per week is far from the recommendation of the WHO [37] and other programs being implemented in other countries [33]. Silva et al. [38] found that engagement in three classes per week has a higher impact among students, which suggests that the SEP in Mexico should look beyond the current political and educational scope to create appropriate spaces to a more effective time of PE, which follow local and international tendencies that let students engage in a quality PE class. 

In our findings, the lack of congruency in the hiring process and schemes was also identified, which was reflected in having classes with numerous students. This is consistent with Galaviz et al. [11] who found a teacher deficit being correlated with the number of students in a class. This is also consistent with Olvera et al. [13] who identified different professional profiles to teach PE and the need of hiring teachers to improve the quality of the classes, which suggest a new opportunity to improve teachers’ recruitment, attributed to the education authorities. In this way, it affects the managing aspect that the school district supervisor handles in every school year. 

With respect to the lack of equipment and the need to improve the facilities, it is a situation that is not only located in a specific region, but is also nationwide, which is consistent with Olvera et al. [13] who found that this aspect creates a management problem that PE teachers face, who have no direct control for the administrative process involved. According to Galaviz et al. [11], at least 70% of schools in Mexico have a specific area to teach PE; however, these spaces are not always used for that purpose, and this is also recurrent around Latin-American countries [29]. Dewi et al. [39] mentioned that schools should have adequate facilities in order to increase the learning potential of students, which is a fundamental purpose of schools. Hanggara et al. [40] said that it is essential to develop educational policies that favor the management and care of the school facilities. Thus, as it was mentioned, the Mexican education authorities should develop infrastructure projects in a different way, although it should be acknowledged that equipment and facility spaces in PE are a collaborative work that involves teachers, students, and administrative leadership.

## 5. Conclusions

This research focused on understanding the factors that influenced a quality PE. We wanted to see these aspects from a holistic point of view and the administrative leadership. What this means is that we wanted to understand the link between these perspectives as a way to provide a more thorough view. In Mexico, the PE class in the basic education does not receive the importance required for international entities. According to our findings, we recognize the government efforts through its leadership, although these efforts seem to be insufficient and incongruent between what is said and done. In this way, it is suggested to establish strict policies and guidelines that could be applicable to not only some regions, but nationwide. This was the purpose of applying different projects and unifying the hiring criteria to increase the PE classes per week, which lets students complete the required amounts of physical activity, and also to manage equipment and facilities. At the same time, it is suggested to create a stronger link between PE higher education institutions and the governing parties as a way to tackle this issue. As such, it can be said that it is necessary to strengthen the education leadership especially in PE, in order to create an opportunity to increase the organization, administration, and intervention that facilitates students’ engagement in physical activity for a lifetime.

## 6. Limitations of the Study

This study’s purpose was to identify the different associated factors that may influence a quality PE in order to contribute to future teachers’ professional development. However, some limitations were as follows: (1) the lack of specific literature in this regard coming from a Mexican perspective; (2) qualitative research can provide an insight, although it can be a subjective one that is only specific to the context; (3) participants came only from the administrative leadership; thus, their views can lack practical sense. Future perspectives can focus on the different factors that teachers hold from their perspective, thus completing the full view of this identification issue.

## Figures and Tables

**Figure 1 ijerph-19-02869-f001:**
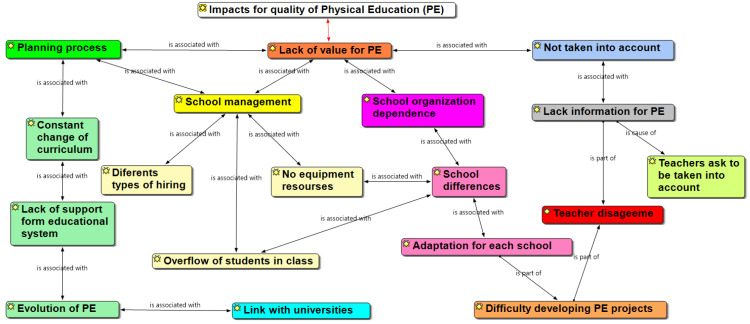
Impact of quality of physical education scheme.

## Data Availability

Some or all data that support the findings of this study are available from the corresponding author upon reasonable request.

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
