# Peer review of "Analysis of the Factors That Influence a Quality Physical Education in Mexico: School Supervision’s Perspective"

_ijerph, 2022, doi:10.3390/ijerph19052869_

Round 1

Reviewer 1 Report

Teaching physical education is an important issue. Especially at a time when physical activity is very limited. The introduction is generally very well written. The Introduction contains all the necessary information. I think pointing out the issue is very important, but the scientific method used for the three participants is very weak, and therefore I do not recommend the publication.

Change interpretation „

 It was a study with a qualitative approach, in which PE area supervisors participated, data collection was carried out through focus groups and collaboratively  analyzed.“

Ethical approval is missing

This is not appropriate in the participants’ section ‘We wanted to understand their thoughts about their. We also wanted to highlight and understand the supervisor’s perspective, because from this point of view there are several educational policies that can be implemented.’

add exclussion criteria

why is  this brief ?    
„• Joel, rural school supervisor, holding a PE bachelors and master’s degree, with 3 years of experience as supervisor. He supervises schools and PE teachers.  

  • Monica, urban school supervisor, holding a PE bachelors and master’s degree, 109 with 6 years as supervisor. She supervises schools and teachers.
  • Esteban, rural school supervisor, holding a PE bachelors and master’s degree, 111 3 years of experience as supervisor. He supervises schools and teachers.“

Author Response

Thank you very much for the feedback provided to our manuscript. We were happy to see that these recommendations would help strengthen the final draft.

  • As for the methodology section of participants we selected this approach as a way to provide an insight from the leadership perspective, since a qualitative methodology focuses in the context and latent meaning (Schneider, 2013). Thus, the number of participants of this project provide just a glimpse of what should be done.
  • In the methods section, the interpretation was changes and cleared, as a way to complete the sense thoroughly.
  • The ethical approval was added to the procedure and data analysis section.
  • Participants sections we re-wrote the specific sentence as a way to provide a better understanding. We also added and extended the inclusion and exclusion criteria.
  • We extended the description of participants, based on their professional experience as PE teacher and school supervisors.

Reviewer 2 Report

Thank you for the opportunity to review this manuscript. 

Recommendations>

Introduction - Please underline the novelty aspect of your study according with previous researches. 

The conclusions should be extended to highlight the main aspects identified in the study.

Author Response

Thank you very much for the feedback provided to our manuscript. We were happy to see that these recommendations would help strengthen the final draft.

  • The recommendations were added as suggested in the introduction and conclusions. Thank you very much.

Reviewer 3 Report

I would like to thank you for the opportunity to review this article and congratulate the authors for this work. For me, as a physical educator, this topic is very important and has a lot of value. I have enjoyed reading this manuscript very much. Below are my suggestions and at the end my recommendation.

This article analyzed the factors that influence the performance of PE teachers, from the perspective of zone supervisors

Title: The title is concrete, representative and indicative of the problem investigated in the manuscript. As a suggestion, the title should provide information and of the group of subjects being studied.

Abstract: The abstract is clear and complies with the general rules for writing a good abstract. However, I would like to see the sample described a little better, indicating the context. I consider this to be the most important section since it will be read more times than even the manuscript itself.

Introduction: As I mentioned, I find this research extremely important in contributing to the field of Physical Education. I do not disagree with the authors' justifications and I read many very good and current arguments. In addition, the context of Physical Education within the Mexican educational system is correctly argued. It is suggested to the authors that based on the stated objective they highlight the research questions that help to conduct the research and discussion based on the findings found in which the study variables, the study population, and the expected result appear.

Material and method.

Research design: Correct.

Participants. Correct, in this same section the characteristics of the subjects included should be described (sample selection criteria (inclusion and exclusion).

Procedures: Indicate the approval number (line 129).

Results:  The rest of the results are correctly shown and are easy to read and simple for a scholar not used to qualitative methodology .

Discussion: Correct. However, it is suggested to incorporate a section on theoretical and/or practical implications to assess the scope of the research. Please give value to this section. It is suggested to include more specifically the limitations of the study.

Conclusions: They are clear and provide an answer to the stated objectives.

I recommend that this manuscript be sent for another round of review after minor revisions.

Author Response

Thank you very much for the feedback provided to our manuscript. We were happy to see that these recommendations would help strengthen the final draft.

  • The title was modified in order to fit the suggestions made to increase the perspective of the project.
  • Abstract: Changes were made in order to provide a stronger insight of the project.
  • Introduction: Changes were made in order to provide a better insight of the purpose of the project at the end of the section.
  • Participants: Inclusion and exclusion criteria was added and extended as suggested.
  • Procedure: Ethical and revision approval was added to the section.
  • Discussion: Practical expectations and connection to the results were completed as suggested. In order to provide a better insight of the project.

Thank you for the suggestion.

Round 2

Reviewer 1 Report

Dear authors, despite the good changes you have made, I do not recommend the article for publication. The main reason is the poor level of methodologies used.

Author Response

Thank you very much for the recommendation.